# Investigation of Components in Roasted Green Tea That Inhibit *Streptococcus mutans* Biofilm Formation

**DOI:** 10.3390/foods12132502

**Published:** 2023-06-27

**Authors:** Iori Goto, Satoshi Saga, Masaki Ichitani, Manami Kimijima, Naoki Narisawa

**Affiliations:** 1Central Research Institute, ITO EN, Ltd., 21 Mekami, Makinohara 421-0516, Shizuoka, Japanm-ichitani@itoen.co.jp (M.I.); 2Department of Food Science and Technology, College of Bioresource Sciences, Nihon University, 1866 Kameino, Fujisawa 252-0880, Kanagawa, Japannarisawa.naoki@nihon-u.ac.jp (N.N.)

**Keywords:** roasted green tea (RGT), *Streptococcus mutans*, biofilm (BF), polyphenol, glucosyltransferase (GTF)

## Abstract

*Streptococcus mutans* form oral biofilms (BFs) and cause dental caries. Roasted green tea (RGT) is prepared by roasting the tea plant, and RGT-specific polyphenols are produced during the roasting process. Catechins, polyphenols in green tea, have BF inhibitory activity against *S. mutans*; therefore, RGT-specific polyphenols are also expected to have this activity. However, there are few reports on the structural and functional properties of RGT. This study aimed to investigate the inhibitory activity of RGT against *S. mutans* BF formation and to investigate the active compounds. RGT extract fractionation and BF inhibitory assay were performed. Strong activity was confirmed in the RGT fractions that had medium-high hydrophobicity, were rich in phenolic hydroxyl groups, and lacked catechins. A peak comprising compounds with molecular weights of 918 (mw918) and 1050 (mw1050) was purified from the fraction. Since BF inhibitory activity was confirmed for this peak, these compounds were considered to be part of the active ingredients. The mw918 polyphenol was detected only in RGT and it was thought to be produced during the roasting process. The results of this research will serve as a basis for the future application of RGT as a safe and effective anti-caries agent.

## 1. Introduction

*Streptococcus mutans* are gram-positive facultative anaerobic bacteria that are a major cause of dental caries [1]. *S. mutans* form biofilms (BFs), known as dental plaque, consisting of an extracellular polysaccharide matrix in the oral cavity [2]. *S. mutans* produce glucosyltransferases (GTFs), such as GtfB and GtfC. GTFs catalyze the production of insoluble glucans, such as mutans, which are the major components of BFs formed by *S. mutans* [3]. The acid tolerance of *S. mutans* in BFs increase due to their acid resistance response [4], and resistance to antibiotics also increases due to the reduced access of antibiotics to these *S. mutans* [5]. Furthermore, BFs adhere tightly to the tooth surface, making it difficult to remove them mechanically [5]. *S. mutans* survives in BFs and produces organic acids that lower the pH in the BFs and attack the enamel and mineral components of dentin, causing dental caries [6,7]. Thus, BFs produced by *S. mutans* are the main factor in dental caries, and inhibition of BF formation can be an important countermeasure against caries [8]. Recently, antimicrobial agents, such as fluoride, chlorhexidine, quaternary ammonium salts, and antimicrobial peptides, have been developed to target the oral bacteria that cause oral disease. However, there are concerns that current drugs have low efficacy and side effects, as well as concerns about the emergence of resistant bacteria [9]. Therefore, the development of safe and effective antimicrobial agents is required.

Green tea is produced from the tea plant (*Camellia sinensis* (L.) O. Kuntze), which is a medicinal plant. Green tea extracts have been reported to have antibacterial activity against various bacteria [10] and BF-inhibitory activity against *S. mutans* [11]. Catechins, such as epigallocatechin gallate, present in green tea, exhibit antibacterial activity and inhibit BF formation by *S. mutans* [12,13]. In addition, green tea has been consumed for many years, and its safety has been confirmed. Its application as a safe and effective anti-caries agent is expected. However, gargling with green tea has been reported to be insufficiently effective in reducing the number of oral bacteria because gargling is often avoided due to its strong bitter taste [14]. Overall, reducing the bitterness and astringency of green tea is an issue worth considering for the prevention of dental caries, especially in children.

Roasted green tea (RGT), also called “Hojicha” in Japan, is a type of tea prepared by roasting the stems and leaves of the tea plant at a high temperature. As RGT is less bitter than green tea and has sweet and aromatic flavors; it is popular among most people regardless of age and gender. In recent years, there has been a growing demand for RGT not only for drinking, but also as a food ingredient for lattes and sweets. Despite the growing demand for RGT, there are few research reports on RGT. Although there are a few reports discussing the changes in constituents due to the roasting process, such as polyphenols [15,16,17] and aroma components [18,19], reports on the structural and functional properties of the constituents are limited. Roasting of green tea causes oligomerization of tea catechins [17] and production of polyphenols specific to RGT [16]. A fraction containing polymerized polyphenols fractionated from oolong tea inhibited the action of GTFs in *S. mutans* [20,21], and polymerized polyphenols of RGT may inhibit BF formation by *S. mutans*. Therefore, RGT is expected to inhibit BF formation by *S. mutans*, and may be more applicable than green tea for this role, owing to its low bitterness and astringency.

In this study, we aimed to evaluate the inhibitory activity of RGT on *S. mutans* BF formation and to identify the active ingredients. This research will serve as a basis for the future application of RGT as a safe and effective anti-caries agent.

## 2. Materials and Methods

### 2.1. Preparation of Crude Extract of RGT and Unrefined Tea (UT)

RGT and UT (also called “Aracha” in Japan), as the raw material of RGT, which is produced in Shizuoka Prefecture in Japan, were purchased (Takeharaen Tea Factory Co., Ltd., Shizuoka, Japan). Dry tea leaves were ground in a food processor; 3 L of 70% ethanol solution was added to 300 g of the ground tea leaves and extracted for 18 h at about 25 °C with stirring. The tea leaves were removed from the extract by centrifugation (3000 rpm, 4 °C, 10 min) and filtration (Advantec No. 1; Advantec Toyo Kaisha, Ltd., Tokyo, Japan), and the supernatant was collected. The supernatant was then concentrated under reduced pressure using a vacuum evaporator (N-1000; Tokyo Rikakikai Co., Ltd., Tokyo, Japan). Finally, the extracted powder was obtained using a vacuum freeze dryer (VD800-R; Taitec Co., Ltd., Saitama, Japan). The dried extracts were named RGT Extract (REx) and UT Extract (UEx).

### 2.2. Fractionation

The fractionation flowchart and correspondence between fractions and abbreviations are shown in Figure 1. The first fractionation based on hydrophobic interaction was performed using DIAION^TM^ HP20 (Mitsubishi Chemical Co. LTD., Tokyo, Japan) (h 22 cm × Φ6.0 cm) with REx and UEx as raw materials. Gradient elution was performed at a flow rate of 30 mL/min in the following order: 3.6 L of 20 *v*/*v*% ethanol solution, 2.4 L of 40 *v*/*v*% ethanol solution, and 2.4 L of 80 *v*/*v*% ethanol solution; three fractions were obtained from each of the extracts (RFr. 1–RFr. 3 and UFr. 1–UFr. 3, respectively) (the fractions of REx and UEx were named RFr and UFr, respectively).

Secondary fractionation based on size exclusion and interaction with phenolic hydroxyl groups was performed using TOYOPEARL^®^ HW-40EC (Tosoh Co. Ltd., Tokyo, Japan) (h 22 cm × Φ3.0 cm) with RFr. 2 or UFr. 2 as raw materials. The gradient elution was performed at a flow rate of 18 mL/min in the following order: 450 mL of 1% acetic acid aqueous solution, 450 mL of 40 *v*/*v*% methanol in 1% acetic acid aqueous solution, 450 mL of 80 *v*/*v*% methanol in 1% acetic acid aqueous solution, and 450 mL of 1% acetic acid methanol solution; four fractions were obtained from each of the extracts (RFr. 4–RFr. 7 and UFr. 4–UFr. 7, respectively).

Preparative chromatography based on hydrophobic interaction was performed using YMC-DispoPackAT ODS-25 (h 10 cm × Φ2.0 cm) with the above RFr. 6 or RFr. 7 as raw materials. Separation was performed at a flow rate of 11 mL/min with the following gradient: 0–35 min, 1% acetic acid aqueous solution; 0–2 min, 10% acetonitrile; 2–3 min, 10–15% acetonitrile; 3–11 min, 15% acetonitrile; 11–22 min, 15–30% acetonitrile; 22–27 min, 30% acetonitrile; 27–30 min, 30–80% acetonitrile; 30–35 min, 80% acetonitrile. Fractionation was performed when all 8 catechins were eluted. Thus, fractions with and without catechins were obtained (fraction with catechins from RFr. 6: RFr. 6-1, fraction without catechins from RFr. 6: RFr. 6-2, fraction with catechins from RFr. 7: RFr. 7-1, and fraction without catechins from RFr. 7: RFr. 7-2).

Molecular weight fractionation was performed according to a previously published method [22] to further investigate the active components of RGT. HW-40C (h 55 cm × Φ2.6 cm) was used with the above RFr. 6-2 and RFr. 7-2 as raw materials for fractionation. The solvent was a mixture of 7 M urea solution and acetone in a ratio of 2:3, and the flow rate was free-fall (approximately 0.5 mL/min). The eluate was collected every 30 min and the distribution of phenolic compounds was confirmed by HPLC (UV, 230 nm). The eluate was divided into three fractions according to the characteristics of the components to facilitate subsequent peak isolation. The fractions were collected in the order of elution (fractions from RFr. 6-2: RFr. 6-2-1 to RFr. 6-2-3, and fractions from RFr. 7-2: RFr. 7-2-1 to RFr. 7-2-3). Urea in the solution was removed with HP20 (h 24 cm × Φ3.5 cm); isocratic elution was performed with water.

Finally, to isolate the components in RGT, peak isolation was performed using RFr. 6-2-2; gradient elution with 1% acetic acid in aqueous methanol solution, at a flow rate of 12 mL/min, was performed using WakoSil-II 5C_18_ HG Prep (h 25 cm × Φ2.0 cm), and three peaks (Peak A, Peak B, and Peak C) were separated (shown in Figure 2).

### 2.3. Measurement of BF Inhibitory Activities

BF inhibitory activities were measured according to the previously reported protocol [23,24] with further modifications. *S. mutans* NBRC13955^T^ was purchased from the Nite Biological Resource Center (NBRC), and the bacterial stock solution was prepared as designated by NBRC. The bacterial solution was cultured in Brain Heart Infusion (BHI) medium (BD BBL™, Nippon Becton Dickinson Company, Ltd., Tokyo, Japan) at 37 °C for 18 h under 5% CO_2_ conditions (AnaeroPack CO_2_; Mitsubishi Gas Chemical Co., Inc., Tokyo, Japan). The bacterial culture was suspended in sterilized water to obtain an optical density of 0.3 at 660 nm. The bacterial suspension was centrifuged (10,000 rpm, 5 min) and washed twice with sterilized water to remove the medium components. Finally, the bacterial cells were suspended in 1 mL of sterilized water (approximately 10^7−8^ CFU/mL).

Each extract or fraction was suspended in water at 1% (*w*/*w*) concentration (50 mg/5 mL), dissolved at 80 °C for 10 min, and filtered (through 0.22 µm, hydrophilic PTFE) for sterilization. The filtered solution was 2-fold-diluted with sterilized water and used for the BF inhibitory assay.

The test solution was inoculated in 0.2% sucrose-added tryptone soya broth without dextrose medium (Merck Biopharma Co., Ltd., Tokyo, Japan) at a final concentration of approximately 10^5−6^ CFU/mL, with the sample at a final concentration of 0.5–0.008%, and incubated at 37 °C for 20 h under 5% CO_2_ conditions to form a BF at the bottom surface of the plate. The formed BF was washed three times with PBS and air-dried for subsequent staining.

The formed BF was then stained with a 1% safranin solution for 15 min, and the excess solution was washed three times with PBS and then air-dried. The safranin remaining in the BF was extracted with 150 µL of 70% EtOH for 30 min, 50 µL of the extract was transferred to a new plate, and the absorbance (Abs_492nm_) was measured with a plate reader (Sunrise Rainbow RC; Tecan Japan Co., Ltd., Kanagawa, Japan). The absorbance represented the amount of BF formed.

The minimum BF inhibitory concentration (MBIC) was determined as the lowest concentration of the tested solution that inhibited BF formation by ≥ 50% (MBIC_50_) compared to that of the untreated control [7,25].

### 2.4. Measurement of GTF Inhibitory Activities

For the preparation of GTF fraction, *S. mutans* UA159 cultures grown in BHI medium were centrifuged at 8000× *g* for 10 min at 5 °C. The supernatant was salted by ammonium sulfate (Fujifilm Wako Pure Chemicals Co., Ltd., Osaka, Japan) precipitation at 60% saturation at 4 °C for 3 h. After salting, centrifugation was performed at 10,000 rpm at 4 °C for 30 min, and the precipitated proteins were dissolved in 0.1 M PBS (pH 7.4). The sample solution was dialyzed with >100-fold sample solution in 0.1 M PBS for 3 h using a cellulose dialysis membrane (molecular weight, 12,000–14,000) (Viskase Companies Inc., Lombard, IL, USA). After dialysis, ultrafiltration was performed on a 100 kDa ultrafiltration filter (Amicon^®^ Ultra-100K, Merck, Darmstadt, Germany). The resulting >100 kDa fraction was suspended in 0.1 M PBS (pH 7.4) and concentrated 60-fold. The glucan production capacity of the crude enzyme solution obtained from the *S. mutans* culture medium was evaluated. Glucan production capacity was defined as the amount of insoluble glucan synthesized by the crude enzyme solution in a reaction with sucrose as substrate at 37 °C for 1 h.

The final concentration was adjusted to 2.0 mg protein/mL crude enzyme solution and 1% sucrose, and RFr. 6-2 and RFr. 7-2 were added at a concentration of 1/1, 1/10, and 1/100 of the MBIC_50_ value [RFr. 6-2: 0.13% (*w*/*w*), 0.013%, 0.0013%; RFr. 7-2: 0.06% (*w*/*w*), 0.006%, 0.0006%]. The reaction was then carried out in a water bath at 37 °C for 1 h. The enzyme was then inactivated by heat treatment at 90 °C for 15 min, centrifuged at 12,700 rpm at 4 °C for 10 min, and the precipitate was washed twice with the same volume of sterile water as that of the initial reaction solution. It was further washed with 99.5% (*v*/*v*) ethanol and dried. The resulting sample was used as the insoluble fraction. The same volume of a 0.5 M sodium hydroxide solution as that of the initial reaction solution was added to the insoluble fraction and incubated at 37 °C for 1 h. It was then centrifuged at 12,700 rpm at 4 °C for 10 min. The amount of insoluble glucan was determined by the phenol-sulfuric acid method and expressed as glucose equivalent.

Data are shown as the mean value of three independent experiments. Statistical significance of differences between the glucan synthesis levels of samples treated and non-treated with RFr. was determined using Student’s *t*-test (* *p* < 0.05, ** *p* < 0.01).

### 2.5. Quantitation of Catechins and Tannins

Each sample was prepared as a 1% aqueous solution, diluted with pure water as appropriate, and passed through a filter (Versaphor, 0.45 µm; Ekikrodisc 13 water system, Wako Pure Chemicals, Osaka, Japan) before HPLC measurement. HPLC was performed using a Waters HPLC system (Waters, Milford, MA, USA). The analytical separation was performed using a YMC J’sphere ODS-H80 column (250 mm × 3.0 mm, i.d.; YMC Co. Ltd., Kyoto, Japan) at 40 °C with mobile phase A (pure water), mobile phase B (acetonitrile), and mobile phase C (1% aqueous phosphoric acid solution) at a flow rate of 0.43 mL/min. The gradient elution procedure was performed with 82.7% (*v*/*v*) A and 7.3% (*v*/*v*) B from 0 to 5 min, 82.7% to 80.5% A and 7.3% to 9.5% B from 5 to 10 min, 80.5% to 76% A and 9.5% to 14% B from 15 to 25 min, 76% to 49% A and 14% to 41% B from 40 to 45 min, and 49% to 82.7% A and 41% to 7.3% B from 55 to 60 min. A UV-vis detector was used to detect catechins at a wavelength of 230 nm.

Tannins were quantified as an equivalent amount of ethyl gallate using a modified iron tartrate method [26,27]. Briefly, 100 mg of ferrous sulfate 7-hydrate and 500 mg of sodium potassium tartrate were dissolved in water, and the volume was adjusted to 100 mL to prepare the iron tartrate reagent. Sodium phosphate buffer was adjusted to pH 7.5 to prepare the Serenzen buffer. Each sample was prepared as a 1% aqueous solution and diluted with pure water, as appropriate. Then, 60 µL of sample solution, 60 µL of iron tartrate reagent, and 180 µL of Serenzen buffer were mixed in each well of a 96-well microplate, and the absorbance was measured at 540 nm using a plate reader. A calibration curve was constructed using an ethyl gallate solution as an external standard, and the amount of tannins in the sample was determined as ethyl gallate equivalents.

### 2.6. Mass Spectrometry (MS)

MS analysis was performed using a Waters Synapt HDMS system (Waters). Analytical separation was performed on a BEH C_18_ column (2.1 mm × 100 mm, i.d.; 1.7 μm, Waters) at 30 °C using mobile phases A (water/formic acid, 99.9: 0.1, *v*/*v*) and B (acetonitrile) at a flow rate of 0.5 mL/min. The gradient elution procedure was conducted with 7% (*v*/*v*) B for 0–2 min, 7% to 25% (*v*/*v*) B for 2–8 min, and 60% (*v*/*v*) B for 9 min.

### 2.7. Nuclear Magnetic Resonance (NMR) Spectroscopy

NMR spectra were recorded on JNM-ECX400Ⅱ (Jeol Resonance Co., Ltd., Tokyo, Japan) with standard pulse sequences operating at 400 MHz for ^1^H NMR of Peak A, Peak B, and Peak C, and at 100 MHz for ^13^C NMR for Peak A.

## 3. Results and Discussion

### 3.1. Quantitative Values of Catechins and Tannins

The results of the quantification of catechins and tannins are shown in Table 1. The total catechin content of the extracts was 13.27 mg/100 mg for REx and 34.62 mg/100 mg for UEx, indicating that the total catechin content decreased by about 1/3 during the roasting process of RGT. The tannin content was 24.8% (*w*/*w*) for REx and 23.9% (*w*/*w*) for UEx, suggesting that tannins other than catechins were produced during the roasting process. The total catechin content of the RGT HP20 fraction (RFr. 1–RFr. 3) was about 1/2–1/3 of that of UFr. Catechin contents of both RFr and UFr, ranked highest to lowest, were Fr. 2, Fr. 1, and Fr. 3. Fr. 1 had more non-gallate catechins, while Fr. 2 had more gallate catechins. For the subsequent HW40-EC fraction (Fr. 4–Fr. 7), the total catechins were lower in RFr than in UFr as before, and both RFr and UFr had more catechins in the order of Fr. 7 and Fr. 6, while Fr. 4 and Fr. 5 had almost no catechins. Most of the non-gallate catechins were fractionated to Fr. 6, and most of the gallate catechins were fractionated to Fr. 7. After the preparative chromatography using ODS-25, almost all of the catechins were successfully distributed to RFr. 6-1 and RFr. 7-1. Almost no catechins were detected in RFr. 6-2 and RFr. 7-2, while approximately 26% (*w*/*w*) of tannins were detected.

### 3.2. MBIC_50_ of the Extracts and Fractions

The yield and MBIC_50_ of the extracts and fractions of RGT and UT are shown in Table 2. The MBIC_50_ values of REx and UEx were 0.25% and 0.50%, respectively, and at these concentrations, the extracts had no effect on the growth of *S. mutans*. This result indicates that RGT exhibits a stronger BF inhibitory activity than UT, and the bacterial growth is not related to the inhibitory activity. The total catechins and EGCg contents of REx were almost one-third of those of UEx (Table 1). Catechins, especially EGCg, inhibit BF formation by *S. mutans* [12,13]. As REx showed a stronger activity despite the decrease in catechins, this suggests that the contents of active compounds other than catechins were higher in RGT.

Separation based on hydrophobic interactions using HP20 revealed that the latter separated fraction (Fr. 2 and Fr. 3) showed strong activity. Therefore, the main active compounds were considered to have moderately high hydrophobicity. The HPLC chromatogram (UV, 230 nm) of RFr showed a peak that was not observed in UFr, suggesting that phenolic compounds unique to RGT are also present in this fraction. Although the MBIC_50_ of both RFr and UFr was 0.13%, the yield of the RFr was 2–3 times higher than that of UFr, suggesting that the amount of active compounds was higher in RGT than in UT. RFr. 2 showed a high yield and strong BF inhibitory activity and was used as the raw material for the next fractionation.

Among the HW-40EC fractions, the formerly separated fractions (Fr. 4 and Fr. 5) were inactive (MBIC_50_ ≥ 0.50%), but the further separated fractions (Fr. 6 and Fr. 7) showed strong activity. Therefore, the main active compounds were considered to contain high levels of phenolic compounds. RFr. 6 and RFr. 7, with a high polyphenol content (35.80% (*w*/*w*) and 51.79% (*w*/*w*) tannin content, respectively, showed stronger activity than UFr. 6 and UFr. 7.

For the ODS-25 fractions, the MBIC_50_ values of RFr. 6-1 and RFr. 6-2 were 0.13% and 0.06%, respectively. The MBIC_50_ values of RFr. 7-1 and RFr. 7-2 were the same (0.03%). Fractions without catechins were more active than, or equal to the fractions containing catechins, indicating that polyphenols other than catechins have strong activity in RGT.

Molecular weight fractionation using HW-40C with urea was performed to isolate the components from the fraction without catechins. A strong BF inhibition was observed for all fractions (RFr. 6-2-1–RFr. 6-2-3 and RFr. 7-2-1– RFr. 7-2-3). Following fractionation and purification, fractions containing polyphenols with medium-high hydrophobicity and rich in phenolic hydroxyl groups showed a strong BF inhibitory activity.

In RFr. 6-2-2, Peak A was considered isolable. After isolation, the MBIC_50_ value of Peak A was 0.25%. This was slightly lower than that of the original fraction; however, as activity was observed, it was considered to be part of the active ingredients. It is also possible that other active components are present in the other fractions. However, as these components are difficult to purify due to the very complex matrix, they could not be isolated in this study, warranting further studies in the future.

### 3.3. Inhibition of GTFs

The amount of glucans produced by GTFs under treatment with RFr. 6-2 or RFr. 7-2 is shown in Figure 3. Glucan production by GTFs was significantly reduced by treatment with RFr. 6-2 or RFr. 7-2 at a concentration of 1/10 of the MBIC_50_ value. The decrease in glucan production by GTFs was considered to be concentration-dependent. As oolong tea polyphenols have been reported to inhibit the enzymatic activity of GTFs [20,21], the polyphenols of RGT are also considered to act on GTFs and inhibit their enzymatic activity. Moreover, tea catechins suppress the expression of *gtf* genes in *S. mutans* [13]. We also confirmed the suppression of *gtfB*, *gtfC*, and *gtfD* expression in *S. mutans* UA159 in the presence of 0.13% RFr-2 or 0.06% RFr. 7-2 (data not shown). This result strongly suggests that RGT-derived components may suppress the expression of *gtf* genes and inhibit their enzymatic activity in *S. mutans*, resulting in the inhibition of BF formation. However, further investigation of the suppression mechanism is required and will be the subject of future studies.

### 3.4. MSMS and NMR

The molecular weights of Peaks A, B, and C were estimated from the results of LC/MS (Figure 4). Peak A was estimated as a mixture of components with molecular weights of 918 (mw918) and 1050 (mw1050). The mw1050 component has been previously reported [28], and a TOF-MS/MS spectra of [M-H]^−^ ion of mw1050 was shown but no [M-H]^−^ ion of mw918 was identified in the spectra. Based on these collective findings and the results of the NMR measurements described below, we considered Peak A to be a mixture of mw1050 and mw918. Peak B was estimated as a component with a molecular weight of 888 (mw888), and Peak C was estimated as a component with a molecular weight of 1034 (mw1034). Components with mw1050, mw888, and mw1034 were confirmed to be present in UT, whereas that with mw918 was not present in UT, suggesting that components with mw918 were specific to RGT.

Next, to estimate the structure of the peaks, an NMR analysis was performed. We attempted to estimate the structure of the peaks from the NMR spectrum, referring to a previously reported method [29].

The NMR spectroscopic data of Peak A are listed here. For ^1^H-NMR (400 MHz, methanol-*d*4) they were: δ 7.68 (d, J = 15.6 Hz, 1H), 7.60 (d, J = 2.3 Hz, 1H), 7.54 (dd, J = 6.6, 1.6 Hz, 1H), 7.46 (d, J = 8.7 Hz, 2H), 6.87 (d, J = 8.2 Hz, 1H), 6.80 (d, J = 8.7 Hz, 2H), 6.40 (d, J = 16.0 Hz, 1H), 6.33 (d, J = 1.8 Hz, 1H), 6.14 (d, J = 2.3 Hz, 1H), 5.55 (d, J = 7.8 Hz, 1H), 5.23 (t, J = 8.7 Hz, 1H), 4.61 (d, J = 1.8 Hz, 1H), 4.45 (d, J = 7.8 Hz, 1H), 4.34 (d, J = 6.9 Hz, 1H), 4.00–3.25 (m, 14H), and 1.12 (d, J = 6.0 Hz, 3H). For ^13^C-NMR (101 MHz, methanol-*d*4) they were: δ 177.7, 167.4, 164.3, 161.7, 159.9, 157.6, 157.1, 148.3, 146.0, 144.5, 133.6, 130.0, 126.0, 124.5, 122.1, 121.9, 116.2, 115.4, 114.8, 113.8, 104.5, 104.2, 104.0, 101.0, 99.6, 98.5, 93.5, 83.1, 81.8, 80.8, 76.1, 75.4, 74.1, 73.2, 72.5, 71.2, 70.9, 69.9, 69.5, 68.8, 68.1, 67.1, 60.7, and 16.6.

The NMR spectroscopic data of Peak C were as follows. ^1^H-NMR (400 MHz, methanol-*d*4) δ 7.96 (d, J = 8.7 Hz, 2H), 7.66 (d, J = 16.0 Hz, 1H), 7.44 (d, J = 8.7 Hz, 2H), 6.87 (d, J = 9.2 Hz, 2H), 6.78 (d, J = 8.7 Hz, 2H), 6.36–6.32 (m, 2H), 6.14 (d, J = 2.3 Hz, 1H), 5.53 (d, J = 7.8 Hz, 1H), 5.16 (t, J = 8.9 Hz, 1H), 4.55 (s, 1H), 4.35 (dd, J = 33.2, 7.1 Hz, 2H), 3.94–3.21 (m, 21H), 1.09 (d, J = 6.0 Hz, 3H).

For Peak A (mw918 and mw1050 compounds), the ^1^H-NMR data showed that the low-field spectra above 6.0 ppm could be attributed to the structure of quercetin and coumaric acid, and the shift values were generally consistent with these structures. For the spectra from 5.5 ppm to 4.5 ppm, the spectral value and the spin splitting were consistent with the low-field-shifted H spectra of the sugars. The spectra from 4.0 to 3.2 ppm were mixed, and no assignment was possible, but the shift values suggested that the protons were derived from glycans. In summary, the ^1^H-NMR data showed that the structure was essentially the same as that reported previously [28].

Next, the ^13^C-NMR data of Peak A were analyzed, and the low-field spectra above 110 ppm were assigned to the structures of quercetin and coumaric acid. The high-field spectra could be roughly assigned; however, there were spectra of unknown origin.

MS data revealed the fragment ions at *m*/*z* 917 and 1049, but the mw1050 compound was found not to produce the fragment ion at *m*/*z* 917 by radical cleavage [28]. Therefore, Peak A was considered to be a mixture of mw1050 and mw918 compounds, and the mw918 compound is presumed to be structurally similar to the mw1050 compound.

Based on these results and the NMR data, it was concluded that the mw1050 compound is a type of quercetin glycoside, which is a combination of quercetin, coumaric acid, and three different sugars (Figure 5). The mw918 compound was thought to be an arabinose-depleted compound of mw1050 (Figure 5).

Both the yield and purity of Peak B (mw888) were considered insufficient compared to those of the other compounds, and the signal integrals were considered to be blurred; therefore, the exact structure of Peak B could not be estimated. However, the low-field signals indicated that it was not a glycoside of quercetin or coumaric acid-like compound A. Based on the NMR and MS spectra, the structure of Peak C (mw1034) was estimated to be equivalent to that of quercetin glycosides (Figure 5). This compound was also suggested to be present in UT.

The mw1050 compound has previously been discovered in tea, and its activity against periodontal disease-causing bacteria has been reported [30]. In this study, the mw1050 compound was detected in both REx and UEx by MS spectrometry. In contrast, the mw918 compound was detected only in RGT. These results suggest that during the roasting process of RGT, components in the raw material UT were converted to polyphenols, such as mw918. These compounds, including mw918 and mw1050, were found to have BF inhibitory activity against *S. mutans*.

### 3.5. Future Directions

In this research, components in RGT were found to inhibit *S. mutans* BF formation and GTF activity, and to potentially repress *gtf* genes; however, the detailed underlying mechanism remains unclear. The initial adhesion of GTFs and glucans produced by cariogenic bacteria to the bacterial surface layer may have been inhibited by the hydrophobic interactions of polyphenols. The highly hydrophobic fraction showed inhibitory activity, suggesting that the inhibition of the initial adhesion by hydrophobic interactions between the bacterial surface and polyphenols may have inhibited BF formation. Further investigations of the effects of these fractions and compounds on GTFs and the metabolism and gene expression of *S. mutans* as well as the suppression mechanism will be the subject of future studies.

Furthermore, the isolation and identification of mw918 and other active compounds in RGT are urgently needed for their application as more active and potent *S. mutans* BF inhibitors and anti-caries agents.

## 4. Conclusions

A strong *S. mutans* BF inhibitory activity and GTF inhibitory activity was confirmed in RGT fractions containing compounds with medium-high hydrophobicity and rich in phenolic hydroxyl groups. These fractions did not contain catechins, which are traditionally considered to be active compounds. Peak A was separated from one of these fractions, and the results of NMR and MSMS spectrum analyses of Peak A indicated that it is a mixture of the compounds mw918 and mw1050. Since Peak A showed BF inhibitory activity, these compounds were considered to be part of the active ingredients. The mw918 compound was detected only in RGT, indicating that during the roasting process of RGT, components in the raw material UT were converted to polyphenols, such as mw918. These active compounds, including mw918 and mw1050, were found to have BF inhibitory activity against *S. mutans*.

This study is the first to report *S. mutans* BF inhibitory activity and GTF inhibitory activity as well as *gtf* gene repressive activity for non-catechin RGT components. We also fractionated and purified the components involved in the activity and estimated their structures. The results of this research will serve as a basis for the future application of RGT as a safe and effective anti-caries agent.

## Figures and Tables

**Figure 1 foods-12-02502-f001:**
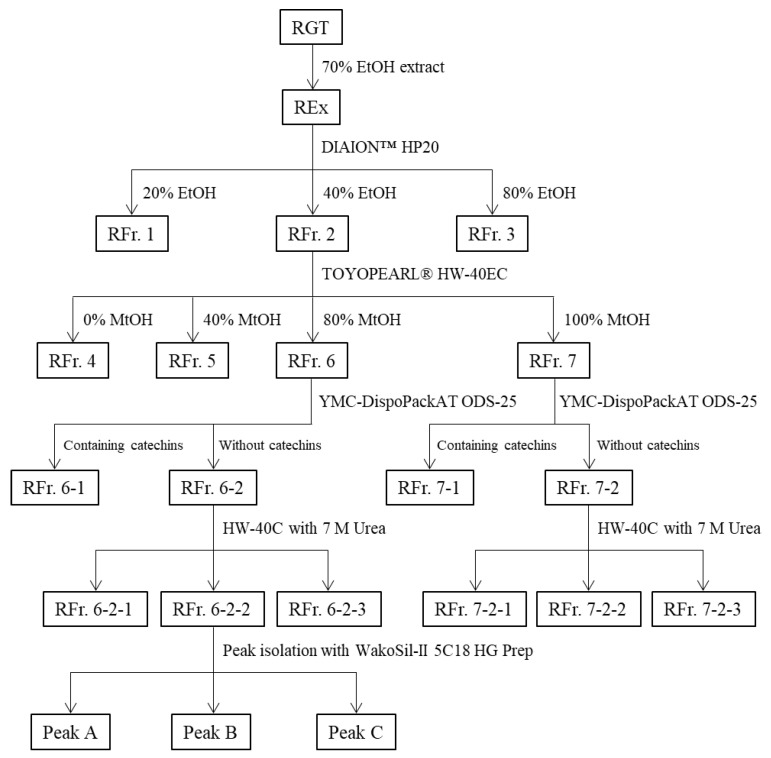
Fractionation flowchart and correspondence between fractions and abbreviations. UEx was also fractionated as shown in this figure.

**Figure 2 foods-12-02502-f002:**
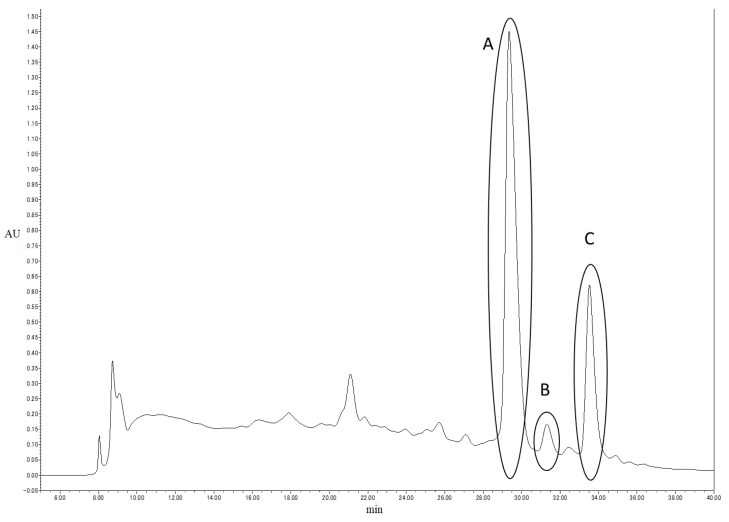
HPLC chromatogram (UV detection, 270 nm) of RFr. 6-2-2. Three peaks were separated and named Peak A, Peak B, and Peak C, respectively.

**Figure 3 foods-12-02502-f003:**
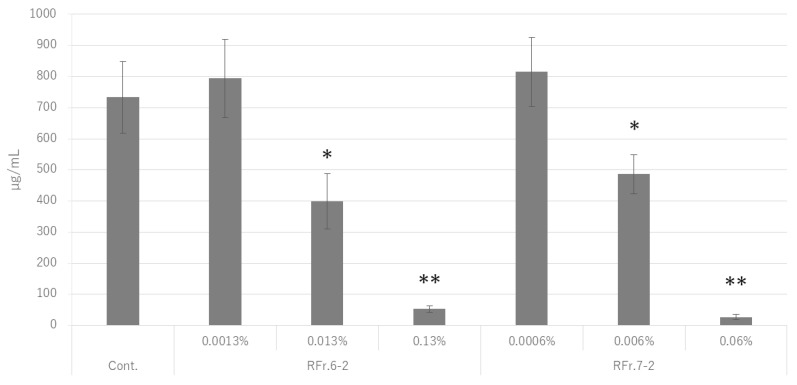
Amount of glucans (µg/mL) produced by GTFs under treatment with RFr. 6-2 or RFr. 7-2. The data were analyzed using Student’s *t*-test (* *p* < 0.05, ** *p* < 0.01). GTFs: glucosyltransferases.

**Figure 4 foods-12-02502-f004:**
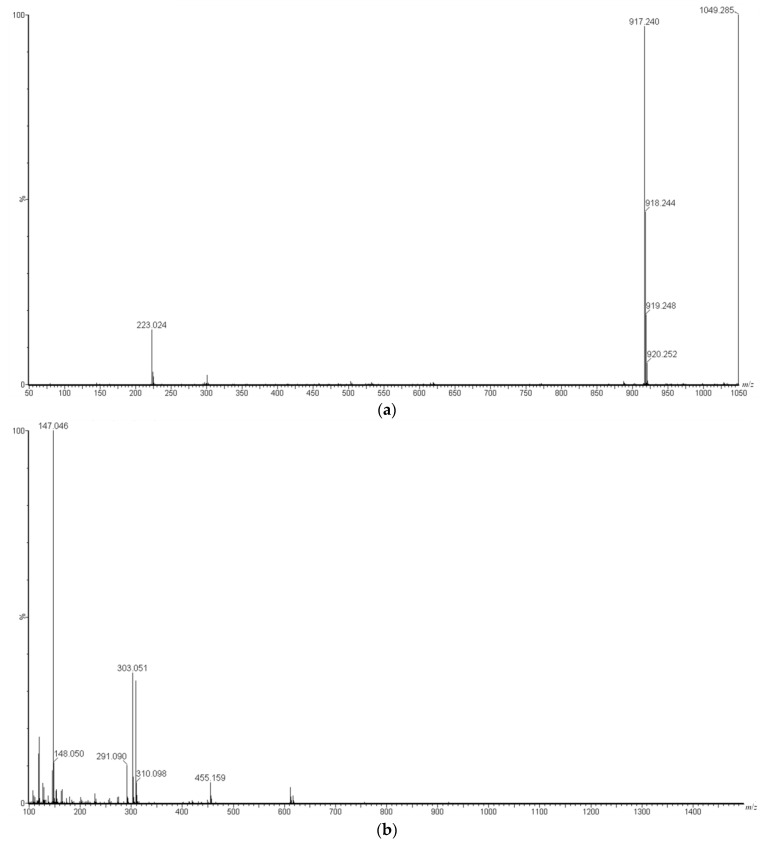
MS data of Peak A. (**a**) MS spectrum; (**b**) MSMS spectrum of mw918; (**c**) MSMS spectrum of mw1050.

**Figure 5 foods-12-02502-f005:**
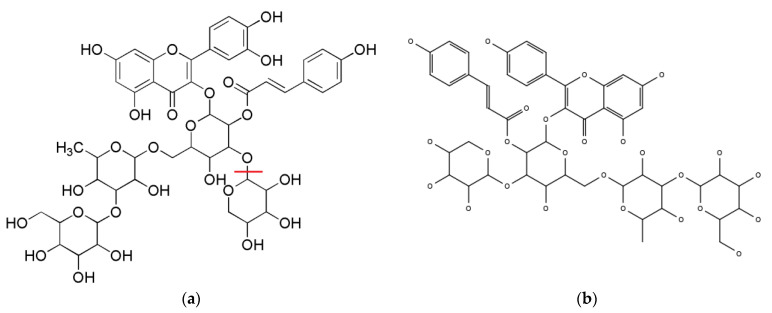
Estimated structures of Peak A and Peak C. (**a**) Estimated structure of Peak A. Mixture of two compounds: mw1050 and mw918. The compound mw1050 was estimated to be a compound with a molecular weight of 1050 and molecular formula of C_47_H_54_O_27_. The compound mw918 was estimated to be a compound in which arabinose was depleted at the red line shown in the figure with a molecular weight of 918 and molecular formula of C_42_H_51_O_23_; (**b**) Estimated structure of Peak C.

**Table 1 foods-12-02502-t001:** Contents of catechins (mg/100 mg) and tannins (% (*w*/*w*); ethyl gallate equivalent).

Fractions	EC	EGC	ECg	EGCg	C	GC	Cg	GCg	Catechins	Tannins
REx	0.73	1.96	0.40	3.68	0.66	2.73	0.86	2.25	13.27	23.9
RFr. 1	0.82	3.07	0.29	4.30	0.93	4.63	LOQ	2.26	16.30	-
RFr. 2	0.78	1.18	2.41	5.27	0.57	1.43	1.35	4.36	17.35	-
RFr. 3	0.49	0.80	0.86	2.37	0.36	0.94	0.30	1.66	7.78	-
RFr. 4	LOQ	LOQ	LOQ	0.06	LOQ	0.14	0.08	0.04	0.32	3.0
RFr. 5	LOQ	0.04	LOQ	LOQ	0.07	0.29	LOQ	0.05	0.45	19.5
RFr. 6	1.91	3.02	LOQ	0.47	1.35	3.50	LOQ	0.07	10.32	35.8
RFr. 7	0.08	0.22	8.75	16.66	LOQ	0.06	4.09	13.92	43.78	51.8
RFr. 6-1	5.54	8.36	2.87	4.12	3.83	8.99	2.79	2.77	39.27	35.9
RFr. 6-2	LOQ	LOQ	LOQ	LOQ	LOQ	LOQ	1.20	LOQ	1.20	26.0
RFr. 7-1	LOQ	LOQ	13.69	29.58	LOQ	LOQ	5.91	26.35	75.54	59.0
RFr. 7-2	LOQ	LOQ	LOQ	LOQ	LOQ	LOQ	LOQ	LOQ	LOQ	26.2
UEx	4.52	15.82	LOQ	10.94	0.31	0.82	2.22	LOQ	34.62	24.8
UFr. 1	4.52	17.50	1.51	10.88	0.30	0.96	LOQ	0.18	35.85	-
UFr. 2	5.41	7.64	10.22	14.13	0.23	0.31	0.18	0.30	38.42	-
UFr. 3	4.52	8.31	3.17	8.82	0.16	0.34	LOQ	0.11	25.43	-
UFr. 4	LOQ	LOQ	LOQ	LOQ	LOQ	0.47	LOQ	LOQ	0.47	-
UFr. 5	LOQ	0.51	0.55	0.62	0.34	LOQ	LOQ	LOQ	2.02	-
UFr. 6	12.58	18.83	7.88	6.84	0.60	0.75	LOQ	LOQ	47.48	-
UFr. 7	0.40	0.54	28.57	45.21	0.34	0.37	LOQ	0.78	76.21	-

EC: Epicatechin, EGC: Epigallocatechin, ECg: Epicatechin gallate, EGCg: Epigallocatechin gallate, C: Catechin, GC: Gallocatechin, Cg: Catechin gallate, GCg: Galocatechin gallate, LOQ: Limit of quantitation, -: Not tested.

**Table 2 foods-12-02502-t002:** Yield (%) and MBIC_50_ (% (*w*/*w*)) of each extract and fraction.

Fractions	Yield	MBIC_50_	Fractions	Yield	MBIC_50_
REx	21.9	0.25	UEx	22.4	0.50
RFr. 1	51.2	0.50	UFr. 1	71.8	0.50
RFr. 2	22.2	0.13	UFr. 2	7.0	0.13
RFr. 3	11.6	0.13	UFr. 3	6.3	0.13
RFr. 4	11.5	>0.50	UFr. 4	18.7	>0.50
RFr. 5	8.9	>0.50	UFr. 5	8.4	>0.50
RFr. 6	16.8	0.06	UFr. 6	33.9	0.13
RFr. 7	10.7	0.03	UFr. 7	20.6	0.06
RFr. 6-1	19.1	0.25			
RFr. 6-2	58.8	0.13			
RFr. 7-1	78.1	0.06			
RFr. 7-2	11.0	0.06			
RFr. 6-2-1	18.5	0.06			
RFr. 6-2-2	22.1	0.13			
RFr. 6-2-3	38.9	0.01			
RFr. 7-2-1	45.5	0.03			
RFr. 7-2-2	32.3	0.02			
RFr. 7-2-3	4.0	0.06			
Peak A	-	0.25			

-: Not calculated, MBIC_50_: Minimum BF inhibitory concentration that inhibited BF formation by ≥ 50%.

## Data Availability

Related data and methods are presented in this paper. Additional inquiries should be addressed to the corresponding author.

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
