# Peer review of "Investigation of Components in Roasted Green Tea That Inhibit Streptococcus mutans Biofilm Formation"

_foods, 2023, doi:10.3390/foods12132502_

Round 1

Reviewer 1 Report

Dear Author, I reviewed the manuscript (foods-2393414) entitled Investigation of components in roasted green tea that inhibit Streptococcus mutans biofilm formation. This manuscript presents relevant information about green tea's anti-biofilm potential against S. mutans. However, some sections of the presented data can be improved. For this reason, I consider that this manuscript needs minor changes to be considered for publication in this journal. 

Additional comments.

Highlight the advantages of using green tea as an antibiofilm treatment against S. mutans.

Check paragraphs extension in this manuscript.

Include an experimental design that contains statistical factors and variables of response in the statistical analyses applied to the findings of this research.

Discuss the green tea catechins antibiofilm mode of action against S. mutans.

Compare the obtained findings with green tea bioactive compounds as antibiofilm treatments. 

Include future trends to keep working with the obtained data. 

Try to conclude with a general statement of the most relevant part of this study.

Author Response

Dear Reviewer,

Thank you for the time and effort you have spent in reviewing our manuscript.

Reviewer 2 Report

The primary objective of this investigation was to examine the inhibitory activity of fractions obtained from RGT against the formation of S. mutans biofilms and to identify the active compounds. The discovery of a potent active compound would significantly contribute to medical development. However, the data provided are insufficient to authenticate the identified compound. Here are my comments:

1、    In the abstract, the authors mentioned the scarcity of reports on the structural and functional properties of RGT. However, the study itself does not present any data regarding the structure and functional properties of RGT.

2、    Firstly, please clarify whether Compound A refers to the compound with a molecular weight of 1050 (mw1050) or not. Compound A should be a pure compound since NMR data for a pure Compound A are provided in lines 296-304.

3、    In Figure 4, the MS spectrum of Compound A, the authors need to clarify whether the two peaks within the spectrum represent a mixture of two compounds or two different ion fragments. The mass spectrometry range should cover m/z >1050.

4、    Throughout the manuscript, the authors repeatedly mention the identification of two compounds with molecular weights of 918 (mw918) and 1050 (mw1050) from RGT. This implies that the peak attributed to Compound A contains a mixture of two compounds, 918 (mw918) and 1050 (mw1050) (in lines 19-20 and 310). They also suggest that the estimated molecular weight of Compound A corresponds to 918 (mw918) and 1050 (mw1050) (in lines 287-288). However, it is possible that Compound A is a pure substance with a molecular weight of 1050 (mw1050), and no identification of two distinct compounds has been made.

5、    If two compounds have indeed been identified, please provide NMR results for mw918 and describe the method used to separate and purify mw918 from the Compound A peak. Additionally, Table 2 should include the mw918 MBIC50 value.

6、    If Compound A is indeed the primary active ingredient, why is its MBIC50 concentration significantly higher compared to other fractions of the mixture (0.03%, 0.06%, or 0.13% vs. 0.25%)?

7、    It is recommended to provide the full names of EC, EGC, and other abbreviations in Table 1. The full names of MBIC50 should be listed in Table 2, and the full names of RTFs should be indicated in Figure 3.

Author Response

Dear Reviewer,

Thank you for the time and effort you have spent in reviewing our manuscript.

We appreciate your insightful and helpful comments.

Reviewer 3 Report

The work is interesting, however difficult to read. For example, I could not understand how you calculated the concentration of the extracts used in the different experiments and as such I could not understand the correlation between the active compounds (catechins and tannins) and antimicrobial activity (BF and GTF inhibition).
other doubts

1-Line 69-RGT and UT (also called “Aracha” in Japan), as the raw material of RGT, which is  produced in Shizuoka Prefecture in Japan -  what exactly is RGT and UT ?   2-Line 71- Tea leaves were ground in a food processor; 3 L of 70% ethanol solution was 71 added to 300 g of the ground tea lea. These tea leaves as they have been processed, are they fresh, dried or roasted?   3-Line 132- The culture 132 medium was suspended in sterilized water to obtain an optical density of 0.3 at 660 nm- Culture medium or bacterial culture? Line 145- The formed BF was washed thrice with PBS and air-dried- twice?   4-Line 175- The same 0.5 M sodium hydroxide solution volume as  the initial reaction solution was added to the insoluble fraction and incubated at 37°C for  1 h. a little confusing sentence   5-In table 1 does LOQ mean not detectable (ND)? And what it meansEC EGC ECg EGCg C GC Cg GCg? this is not my field of study and I am not familiar with these abbreviations   6- Line 272- Glucan production by GTFs was significantly reduced by treatment with RFr. 6-2 or RFr. 7-2 at an MBIC50 value of 0.1%.- nd this value corresponds to what concentration? and what kind of concentration? w/w or concentration of phenolic compounds?   7-Line 279- Based on these, the polyphenols of RGT were suggested to reduce gene expression of 279 GTFs or inhibit their enzymatic activity in S. mutans, resulting in the inhibition of BF for- 280 mation.
this test was not carried out in vitro, that is, you isolated the GTF from a culture of s. mutams and only after extraction and purification of the enzyme was the extract added. If this was the procedure, how might the extract affect gene expression?

Author Response

Dear Reviewer,

Thank you for the time and effort you have spent in reviewing our manuscript.

We appreciate your insightful and helpful comments.

We apologize for the inconvenience about the concentration of samples.

All concentrations of extracts and fractions are unified as w/w% concentration. 

Round 2

Reviewer 2 Report

Although this study is interesting, the current version of the results does not support its conclusion. The study did not truly identify two compounds with molecular weights of MW1050 and MW918 from the roasted green tea that exhibit BF inhibitory activity against S. mutans. Here are my comments:

1.      HPLC analysis revealed three peaks in the RFr. 6-2-2. In my opinion, labeling them as compounds A, B, and C implies that each peak represents a pure compound. However, since peak A is potentially a mixture of two components, it should be separated and purified into two distinct compounds. Then, the authors could precisely name them as compound A, B, C, and D.

2.      The authors should provide mass spectrometry and NMR spectra of the pure compound for meaningful characterization. Currently, the provided mass spectrometry and NMR of the mixture in peak A offer information almost equivalent to a crude fractionation, rather than individual compound characterization.

3.      In Table 2, compound A should be corrected to peak A. However, the table still lacks data for at least the compound with MW918.

4.      The BF assay is the most crucial indicator in this study, yet the authors attribute the lower inhibitory rate of peak A compared to other crude fractionations to the use of protein-rich TSB as the medium. However, the impact of protein in TSB on the other groups is not addressed. In my opinion, if the two components in peak A are the main active substances, their inhibitory rates should have been better than the other crude fractionations. If not, it indicates that the assay may not be suitable. In addition, TSB should be fully spelled out.

5.      Overall, since the results suggest that peak A may be a mixture of two components, the authors should purify the two compounds, analyze their BF inhibitory ability, and determine their structures using mass spectrometry and NMR. Alternatively, the conclusions should be rewritten based on the results obtained, with appropriate logic to revise the entire manuscript.

Author Response

Dear Reviewer,

Thank you for the time and effort you have spent in reviewing our manuscript. We appreciate your insightful and helpful comments. We have provided point-by-point responses to all of your comments. Please see the attachment.

Reviewer 3 Report

the authors responded to my comments and even improved the article

Author Response

Dear Reviewer,

Thank you for the time and effort you have spent in reviewing our manuscript. We are also glad to hear that the manuscript has been improved following our changes based on your suggestions.